# Prevalence of asymptomatic malaria infection by microscopy and its determinants among residents of Ido-Ekiti, Southwestern Nigeria

Azeez Oyemomi Ibrahim[1,2], Ibrahim Sebutu Bello[3]*, Adewumi Oluwaserimi Ajetunmobi[1], Abayomi Ayodapo[4], Babatunde Adeola Afolabi[5], Makinde Adebayo Adeniyi[6]

1 Department of Family Medicine, Federal Teaching Hospital, Ido-Ekiti, Ekiti State, Nigeria, 2 Department of Family Medicine, College of Medicine and Health Sciences, Afe Babalola University, Ad Ekiti, Ekiti State, Nigeria, 3 Department of Family Medicine, Obafemi Awolowo University Teaching Hospitals Complex, Ile-Ife, Osun State, Nigeria, 4 Department of Family Medicine, University College Hospital, Ibadan, Nigeria, 5 Department of Family Medicine, Ladoke Akintola University Teaching Hospital, Oshogbo, Osun State, Nigeria, 6 Department of Community Medicine, Federal Medical Centre, Abeokuta, Ogun State, Nigeria

* bello.ibrahim@gmail.com

**Data Availability Statement:** The dataset is in the Dryad repository and can be accessed with the

## Abstract

### Background

Asymptomatic malaria infections have received less attention than symptomatic malaria infections in major studies. Few epidemiological studies on asymptomatic malaria infections have often focused on pregnant women and children under-five years of age as the most vulnerable groups. However, there is limitation on data regarding asymptomatic infections among the old adult populations, particularly in the study area. Therefore, this study determined the prevalence of asymptomatic malaria infection by microscopy and its determinants among residents of Ido- Ekiti, Southwestern Nigeria.

### Methods

A hospital-based cross-sectional study was conducted between July and September 2021 among 232 consenting apparently healthy individuals aged 40 years and above who were recruited during a free health screening program using a standardised interviewer-administered questionnaire. The questionnaire sought information on respondents' socio-demographics, presence and types of co-morbidity, and the prevention methods being adopted against malaria infection. Venous blood samples were collected and processed for asymptomatic infections using Giemsa-stained blood smear microscopy. Data were analysed using SPSS version 21. Multivariate logistic regression was used to identify factors associated with asymptomatic infections.

### Results

Of the total 232 respondents, 19.0% (48/232) were confirmed to be infected with *Plasmodium falciparum* (95% confidence interval (CI): 14.1% - 24.6%). Lack of formal education (Adjusted odds ratio (AOR): 5.298, 95% (CI): 2.184-13.997), being diabetic (AOR: 4.681, 95% CI: 1.669-

following DOI: https://doi.org/10.5061/dryad.gqnk98ssx.

**Funding:** The authors received no specific funding for this work.

**Competing interests:** The authors have declared that no competing interests exist.

16.105), and not sleeping under Long Lasting Insecticide Net (LLINs) (AOR: 4.594, 95% CI: 1.194-14.091), were the determinants of asymptomatic *Plasmodium falciparum* infection.

## Conclusion

The prevalence of asymptomatic *Plasmodium falciparum* was 19%. Lack of formal education, being diabetic, and not sleeping under LLINs were the determinants of asymptomatic infections.

## Introduction

Asymptomatic malaria infections remain a global public health problem with consequences for human health, as well as social and economic development [1]. Asymptomatic malaria infections refer to the presence of asexual parasites in the peripheral blood without symptoms [2]. Individuals with asymptomatic malaria infections, especially in the adult population have been identified as providing a reservoir for malaria transmission and a precursor in the progression to symptomatic disease [3]. The infections are highly prevalent in endemic areas of Sub-Sahara Africa (SSA), with only a small percentage of individuals exhibiting clinical symptoms [3]. This has grave consequences for malaria control programs [4]. Hence, Asymptomatic malaria infections are recognised as an important obstacle to malaria elimination as it plays a critical role in delaying the global elimination of malaria [5, 6].

Previous studies have shown that after exposure to malaria parasites, a clinical immunity only partially develops depending on some variations such as endemicity, mode of prevention adopted and co-occurrence of co-morbid conditions [7, 8]. Other studies have suggested that increased poverty levels, poor access to quality health care, and unlimited exposure to mosquito bites in rural settings threaten the integrity and sustenance of such partial immunity, especially in the old age categories [9, 10]. It is because of these reasons that advocacy for malaria prevention strategies such as the use of Long-Lasting Insecticide Treated Nets (LLINs), Indoor Residual Spray (IRS), and other measures were introduced to reduce the incidence of asymptomatic infections, especially in rural settings [9]. To eradicate malaria, researchers have advocated for screening individuals for asymptomatic infections using highly sensitive and accurate diagnostic tools. The use of sustained integrated vector control efforts and drug treatment in areas of high-risk populations have also been advocated [11, 12].

Asymptomatic malaria infections have received less attention than symptomatic malaria infections in major studies [5, 7]. Few epidemiological studies on asymptomatic malaria infections have often focused on pregnant women and children under five as the most vulnerable groups [6, 13]. To the best of our knowledge, there has been no published study to attest to the epidemiological data regarding asymptomatic malaria infections among the old adult populations. Yet, the old adult populations are vulnerable to diseases and illnesses due to their physiological state [2, 9, 14]. This is very important because the economic implications of these illnesses on the old adult populations would generally reflect on the well-being of the families, communities, and nations [9]. Thus, the present study determined the prevalence of asymptomatic malaria infections by microscopy and its determinants among residents of Ido-Ekiti, Southwestern Nigeria.

## Materials and methods

### Study area/design/period

A hospital-based cross-sectional study was carried out at the family medicine clinic, federal teaching hospital, Ido-Ekiti, between July and September 2021 among apparently healthy

residents of Ido-Ekiti, Southwestern Nigeria. The study was conducted during the annual free health programs organised for the residents of the community by the hospital chapter of the society of family physicians of Nigeria. The theme of the programs was "screening for communicable and non-communicable diseases among residents of Ido-Ekiti" The hospital is situated at a location central to the host community. Ido-Ekiti is a rural community in Ekiti State. It has a total land area of 332 km2 and a total population of 159,114 inhabitants, according to the most recent population census conducted in 2006. The annual population growth rate is 3.2%, with the population in 2019 estimated to be 225,305 inhabitants [15]. Ido-Ekiti is located in a tropical rainforest with climatic and environmental conditions that support the growth of malaria transmission. Residents of Ido-Ekiti are mainly farmers and traders in the informal sector, with a relatively small proportion comprising the working population and retirees in the formal sector [15]. In the study area, malaria transmission is perennial during the wet season (April – October), with *Plasmodium falciparum* being the major causative agent. The family medicine clinic offers primary and specialist care for a wide array of acute and chronic medical conditions to individuals in its catchment and the surrounding area.

**Source population.** This comprised all adult individuals who were residents in Ido-Ekiti, and who presented for the annual free health programs organised by the society of family physicians of Nigeria.

**Study population.** The study population comprised apparently healthy individuals aged 40 years and above.

**Inclusion/Exclusion criteria.** Consented apparently healthy individuals aged 40 years and above, permanent residents in the host community, have no travel history for at least a week, axillary temperature $\leq 37.5°c$, showed no recent history of fever in the past 48 hours, and any other clinical symptoms of malaria such as headache, dizziness, joint pain, anorexia and malaise were included in the study. Individuals who were too ill that required immediate attention, pregnant women, or those on treatment for malaria or had just completed anti-malaria treatment within two weeks before the conduct of this study were excluded.

**Sample size determination.** This was determined using the formula [16], $n = Z^2 P(1-P)/d^2$ with a prevalence (P) of 16.2% [2] of asymptomatic *Plasmodium falciparum* infection reported in a cross-sectional study among the older population in Southwestern Nigeria, at 95% confidence interval (CI) and 5% margin of error. In this calculation, Z = 1.96, P = 0.162, 1-P = 0.838, and d = 0.05. This gave a minimum sample size (n) of 209. This was increased to 232 to cover dropouts.

**Sampling method.** Systematic random sampling technique was used in this study. The sampling frame is the total number of asymptomatic individuals expected during the study period. Data from the records department for 2018 free health screening among apparently healthy individuals gave a sampling frame of 950. Dividing the sampling frame by the sample size (232) gave a sampling interval of four (4). On each screening day, the fourth registered respondent was selected by systematic random sampling technique. After that, every fourth respondent was selected. This process was repeated on each clinic day throughout the study period until the sample size of 232 was attained. Recruited individuals had their record cards tagged to prevent re-enrolment. The recruitment of the subjects was done by three trained resident doctors in the department who served as research assistants while the researcher did the collection of data.

**Study protocol.** The study instrument was translated from the English language to Yoruba by professional linguists. The process involved forward and back translation. Both the English and Yoruba versions of the questionnaires were used during the study, depending on the language preference of the subject. Pre-testing of the questionnaire was done in a similar

group of subjects at the family medicine department of a nearby sister health facility. Adjustments were made to the study instrument based on evidence from the pre-test.

**Data collection instruments.** The two instruments for data collection were the standardised interviewer-administered questionnaire and the data collection form. The questionnaire sought information about the respondents' socio-demographic characteristics (such as age, gender, education, occupation, and location), mode of malaria prevention adopted by them, and their past medical history. They were also assessed for the presence of co-morbid conditions such as hypertension, diabetes mellitus, Human Immunodeficiency Virus/ Acquired Immune Deficiency Syndrome (HIV/AIDS), Chronic Obstructive Pulmonary Disease (COPD) and Heart failure. These were self-reported.

## Clinical parameters of the respondents

**Microscopy for asymptomatic infections.** Capillary blood samples were collected by finger pricking using a disposable lancet. The thick and thin films were made from the blood sample. The thick and thin smears were prepared on clean, dry microscope glass slides and were allowed to dry. The thin smear was fixed in methanol, and both smears were stained with 5% Giemsa. The stained slides were taken to the hospital laboratory, where parasitological examinations were made independently by two malaria microscopists, with discrepancies resolved by a senior microscopist who ensured quality control. A slide was declared negative if parasites were absent after examining 200 high-power fields. Parasite density was quantified against 200 leucocytes on an assumed leucocyte count of 8000 per ul of blood [17]. The degree of parasite density was graded as mild, moderate, and severe when the counts were between 1-999 parasites/ul, 1000-9999/ul, and > 10,000/ul, respectively, following the method described elsewhere [18].

Parasites/ul = **No.** of asexual stages x 8000 leukocytes/200 leukocytes

**Determination of packed cell volume (PCV).** A micro-haematocrit tube was filled with blood and centrifuged in a micro-haematocrit rotor at 10,000 rpm for 5 minutes. PCV was read using the micro-haematocrit reader and recorded as no anaemia (PCV≥30%), mild anaemia (25-29%), moderate anaemia (20-24%), and severe anaemia (<20%) [19].

**Ethical consideration.** The study protocol was approved by the Ethics and Research Committee of Federal Teaching Hospital, Ido-Ekiti, Ekiti State, Nigeria (ERC/2021/06/25/605A). When seeking consent from the respondents, the methods and objectives of the study were explained clearly to the respondents individually. The respondents were told they were free to refuse or disengage participation at any time without losing any benefit of care or favour to those who participated. Thus, informed written consent was obtained from each respondent before starting the study. Data collected from each respondent and results of laboratory tests were kept confidential, and privacy was ensured throughout the study. The study was at no cost to the respondents.

**Treatment of the respondents.** The results of the laboratory tests were addressed to the respondents, and all malaria parasite asymptomatic individuals were treated in accordance with the treatment guideline for individuals with asymptomatic infections [20]. The reporting of this study conforms to the strengthening of the Reporting of observational studies in Epidemiology (STROBE) statement [21].

## Operational definitions

**Asymptomatic infections.** This is the presence of an asexual parasite in the peripheral blood, in the absence of fever or other acute symptoms, in individuals who have not received recent anti-malarial treatment [22].

**Old adult.** This is referred to as 40 years of age and above in this study and is classified as middle-aged (40-59) and elderly (≥60 years of age).

**Window net.** This is a physical barrier that is placed across the window in a building to prevent Anopheles mosquitoes from entering the room.

**Statistical analysis.** Data were coded, cleaned, entered, and analysed using IBM SPSS for window version 21.0 (IBM Corp., Armonk, NY, USA), respectively. Quantitative data were expressed as mean ± standard deviation. Frequencies were used to determine the respondents' prevalence of asymptomatic *Plasmodium* infection. Binary logistic regression was employed to assess the determinants of asymptomatic *Plasmodium* infection. Variables significant at P-value < 0.05 in the univariate logistic regression were selected for multivariate logistic regression analysis model. Odds ratios with 95% confidence intervals were calculated and P-value< 0.05 was considered statistically significant.

# Results

## Socio-demographic characteristics of the respondents

A total of 232 respondents were studied. The age of the respondents ranged from 40- 83 years with a mean age of *59.3 ± 12.7* years, the majority (59.5%) of them between 41-60 years. The majority of the respondents (65.5%) were males, and about 32.0% of them completed their tertiary education. The majority (68.1%) were rural dwellers, and (71.6%) were of lower income earners (Table 1).

**Table 1. Socio-demographic characteristics of respondents (N=232).**

| Variables | Frequency | Percentage |
|---|---|---|
| **Age (In years)** | | |
| 40-59 (middle-aged) | 134 | 57.8 |
| ≥ 60 (elderly) | 98 | 42.2 |
| Mean age ± | 59.3±12.7 | |
| **Sex** | | |
| Made | 152 | 65.5 |
| Female | 80 | 34.5 |
| **Occupation** | | |
| Farmer | 33 | 14.2 |
| Artisan | 41 | 17.7 |
| Trader | 70 | 30.2 |
| Civil Servant | 72 | 31.0 |
| Retirees | 16 | 6.9 |
| **Education** | | |
| Primary | 21 | 9.1 |
| Secondary | 75 | 32.3 |
| Tertiary | 60 | 25.9 |
| **Domicile** | | |
| Rural | 158 | 68.1 |
| Urban | 74 | 31.9 |
| **Income** | | |
| Low (<27,101Naira) | 166 | 71.6 |
| High (≥ 27,101Naira) | 66 | 28.4 |

27,101 Naira is the monthly equivalent of $2.2 per day at $1 equals to 410.63 Naira as at July 2021.

## Prevalence and parasite density among asymptomatic infections by respondents

In all, 19% (44/232) of the respondents were positive for asymptomatic infections (95% CI: 14.1% - 24.6%), with only *P. falciparum* species identified from thin blood smear. Of the 44 diagnosed with asymptomatic *Plasmodium* infection, 28 (12.1%), 14 (6.0%), and 2 (0.9%) had mild, moderate, and severe parasitaemia respectively (Table 2).

Using multivariate logistic regression for factors associated with asymptomatic *P.falciparum* infection in this study, after adjusting for possible confounders; the odds of being infected with *P. falciparum* was 5.298 times (95% CI: 2.184-13.997) higher among the respondents who had no formal education, 4.68 times (95%CI: 1.669-16.105) higher among the respondents who were diabetes, and 4.594times (95% CI: 1.194-14.091) higher among the respondents who were not sleeping under LLINs (Table 3).

## Discussion

The study identified the prevalence of asymptomatic *Plasmodium falciparum* infection and its determinants among individuals 40 years of age and above, resident in Ido-Ekiti, Southwestern Nigeria. An overall prevalence of asymptomatic *P. falciparum* infection in this study was 19.0% (95% CI: 14.1% - 24.6%), which is similar to 16.2% reported in a cross-sectional study among the asymptomatic elderly population in Oshogbo, Southwestern Nigeria [2]. This may be due to the similarity in the study population, geographical location, climatic and environmental factors. However, a higher prevalence of 50.0% asymptomatic *P. falciparum* infection was reported in a cross-sectional study conducted in Ilorin, North-central Nigeria [23], while another cross-sectional study in Akure Southwestern Nigeria by Adepeju et al. had reported prevalence of 53.3% asymptomatic P. falciparum [24]. While our own study was among the respondents categorised as middle-aged and old adults, the other studies were among the respondents categorised as young adults. This difference agrees with the finding of another study that the prevalence of asymptomatic *P. falciparum* falls as age increases [25]. On the contrary, another cross-sectional study conducted on the prevalence of asymptomatic *P. falciparum* infection in a rural district in Gabonese adults had reported a prevalence of 12% [26], which was lower than the finding in this study. The observed difference in the prevalence of our study compared to these other studies may be due to variations in the study design, study period, and climatic factors. This is because climate variations among study areas have an effect on the life cycle of parasites and parasite adaptation among *Plasmodium falciparum* infections.

The current study has identified factors associated with asymptomatic infections. The respondents who had no formal education were 5.298 times more likely to have asymptomatic

**Table 2. Prevalence and parasite density of asymptomatic infections by Microscopy (N = 232).**

| Variable | Frequency | Percentage |
|---|---|---|
| | **N = 232** | **(%)** |
| **Asymptomatic infections by density** | | |
| Mild parasitaemia ($<$ 1000 parasites) | 28 | 12.1 |
| Moderate parasitaemia (1000 – 9999) | 14 | 6.0 |
| Severe parasitaemia ($\geq$ 10000) | 2 | 0.9 |
| **Prevalence** | | |
| Positive | 44 | 19.0 |
| **95% Confidence Interval (Lower – Upper)** | 14.1% - 24.6% | |

**Table 3. Multivariate logistic regression for the significant factors associated with asymptomatic infections.**

| Variable | +ve (%) | -ve (%) | COR (95% CI) | p-value | AOR (95% CI) | p-value |
|---|---|---|---|---|---|---|
| **Age (in years)** | | | | | | |
| 40 – 59 (ref) | 18(13.4) | 116(86.6) | 1.000 | | 1.000 | |
| ≥60 | 26(26.5) | 72(73.5) | 2.327 (1.192 – 4.544) | 0.012 | 1.975 (0.491 – 5.395) | 0.290 |
| **Occupation** | | | | | | |
| Farmer | 13(39.4) | 20(60.6) | 4.550 (0.884 – 23.407) | 0.055 | 6.123 (0.228 – 16.693) | 0.281 |
| Artisan | 8(19.5) | 33(80.5) | 1.697 (0.319 – 9.022) | 0.532 | 4.259 (0.179 – 10.388) | 0.370 |
| Trader | 12(17.1) | 58(82.9) | 1.448 (0.290 – 7.222) | 0.650 | 2.776 (0.116 – 6.529) | 0.529 |
| Civil Servant | 9(12.5) | 63(87.5) | 1.000 (0.194 – 5.145) | 1.000 | 0.589 (0.026 – 13.587) | 0.741 |
| Retiree(ref) | 2(12.5) | 14(87.5) | 1.000 | | 1.000 | |
| **Education** | | | | | | |
| None | 13(61.9) | 8(38.1) | 16.018 (4.948 – 51.858) | **<0.001** | 5.298 (2.184 – 13.997) | **0.007** |
| Primary | 15(20.0) | 60(80.0) | 2.464 (0.0942 – 6.446) | 0.060 | 3.878 (0.445 – 7.791) | 0.220 |
| Secondary | 9(15.0) | 51(85.0) | 1.732 (0.608 – 4.981) | 0.298 | 2.189 (0.239 – 10.027) | 0.488 |
| Tertiary(ref) | 7(9.2) | 69(90.8) | 1.000 | | 1.000 | |
| **Domicile** | | | | | | |
| Rural(ref) | 24(15.2) | 134(84.8) | 1.000 | | 1.000 | |
| Urban | 20(27.0) | 54(73.0) | 2.068 (1.056 – 4.050) | **0.032** | 2.883 (0.889 – 9.349) | 0.078 |
| **Diabetes Mellitus** | | | | | | |
| Yes(ref) | 22(78.6) | 6 (21.4) | 30.333 (11.148 – 87.495) | **<0.001** | 4.681 (1.669 – 16.105) | **<0.001** |
| No | | | 1.000 | | 1.000 | |
| **Insecticide treated net** | | | | | | |
| Yes(ref) | | | 1.000 | | 1.000 | |
| No | 43(23.9) | 137(76.1) | 16.007 (2.148 – 119.295) | **<0.001** | 4.594 (1.194 – 14.091) | **0.033** |

ref – reference category  COR – Crude Odd Ratio  AOR – Adjusted Odd Ratio.

infections compared to other respondents who had formal education. The result agrees with the reports of other studies that showed that the level of education significantly influences people's knowledge, attitude, and practices, which in turn could lead to reduced malaria infection [27, 28]. However, other studies have reported that people can be acquainted with the knowledge of malaria transmission, prevention, and control irrespective of their educational status [29, 30].

Similarly, the respondents with T2DM were at increased odds of asymptomatic *P. falciparum* infection compared to other co-morbidities. This was in agreement with a previous cross-sectional study conducted in Lagos, which reported that type 2 diabetic individuals harboured asymptomatic *P. falciparum* infection [14]. Furthermore, the finding in the current study was also in agreement with another cross-sectional study conducted in rural Southwestern Nigeria where it was reported that individuals with type 2 diabetes mellitus were associated with the presence of asymptomatic *P. falciparum* [31]. The findings in the current study and these other studies suggest that adult-type diabetic individuals were potential reservoirs of asymptomatic infections. The mechanisms of occurrence of asymptomatic malaria infections in adult T2DM are not completely understood. However, a recent study from central Africa has attributed the association to anti-parasitic immunity, which prevents parasitaemia from reaching a clinical level called pyrogenic threshold and for preventing secondary infection in adults with primary asymptomatic infections [32]. Other studies have linked the association between asymptomatic infections and individuals with T2DM to increased mosquito bites, and olfactory signals mediate mosquito attraction [14, 31, 33]. The current finding calls for the

investigation of high-risk individuals aged 40 years and above for asymptomatic carriage of malaria parasites as an intervention for ensuring malaria elimination. Contrary to our findings, other authors have reported symptomatic infections among individuals with T2DM [9, 34].

In the same vein, using LLINs was found to be protective against asymptomatic infections as the data showed that respondents who did not sleep under LLINs were five times more likely to have asymptomatic infections as compared to respondents who slept under LLINs. This finding agrees with a previous cross-sectional study conducted among the elderly population in Oshogbo, where it was reported that individuals who did not sleep under LLINs were at increased risk of asymptomatic infections [2]. Appropriate utilisation of LLINs is one of the major cost-effective interventions for malaria prevention [23]. This finding calls for a continuous awareness campaign on the usefulness and usage of LLINs in rural settings. However, a cross-sectional study conducted in Abuja, Nigeria by Onyiah et al., had found no association between individual LLIN use and asymptomatic infections [35].

## Limitations

Firstly, the study was based on a cross-sectional design and, thus, had limited opportunities to measure any causal association between asymptomatic infections and other factors. Microscopy instead of polymerase chain reaction (PCR) was used to diagnose asymptomatic infections in this study. We may therefore have underestimated the burden of asymptomatic infections given the fact that PCR sensitivity can extend to below one parasite per microlitre [36]. However, PCR was not used in this study due to non-availability in our facility. Also, the information provided on preventive methods was self-reported and could not be verified. Therefore, ultimate corroboration would need a prospective longitudinal study controlling for exposure. Despite these limitations, the study generates vital information regarding the burden and the associated risk factors for asymptomatic infections, which could be helpful in the formulation of further steps to implement control interventions.

## Conclusion

In this study, the prevalence of asymptomatic *P. falciparum* infection was 19.0%. The respondents with informal education, diabetes mellitus and those not sleeping under LLINs were identified as determinants of asymptomatic infections. Strategies promoting LLINs use need to be intensified and should be complemented by awareness campaigns against asymptomatic infections in Ido- Ekiti.

## Supporting information

**S1 File. Dryad data set.**
(PDF)

**S1 Data. ASM data.**
(XLSX)

## Acknowledgments

The authors expressed profound gratitude to the hospital chapter of the society of family physicians for the free health programs, which provided an avenue to conduct this research. Appreciation goes to the resident doctors and nurses in the family medicine department. The authors also thank our respondents for making themselves available for the research.

## Author Contributions

**Conceptualization:** Azeez Oyemomi Ibrahim.

**Data curation:** Azeez Oyemomi Ibrahim.

**Formal analysis:** Azeez Oyemomi Ibrahim, Adewumi Oluwaserimi Ajetunmobi, Abayomi Ayodapo, Makinde Adebayo Adeniyi.

**Investigation:** Azeez Oyemomi Ibrahim.

**Methodology:** Ibrahim Sebutu Bello, Adewumi Oluwaserimi Ajetunmobi, Abayomi Ayodapo, Babatunde Adeola Afolabi, Makinde Adebayo Adeniyi.

**Project administration:** Adewumi Oluwaserimi Ajetunmobi.

**Supervision:** Ibrahim Sebutu Bello, Abayomi Ayodapo, Babatunde Adeola Afolabi.

**Writing – original draft:** Azeez Oyemomi Ibrahim.

**Writing – review & editing:** Ibrahim Sebutu Bello, Adewumi Oluwaserimi Ajetunmobi, Abayomi Ayodapo, Babatunde Adeola Afolabi, Makinde Adebayo Adeniyi.

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
