## [Decision Letter · Decision Letter 0]

26 Sep 2022

PONE-D-22-24029Socio-Demographic Profiles, Co-Morbid Conditions and Preventive Measures as Determinants of Asymptomatic Malaria Infections Among Older Age Population in Rural Southwest NigeriaPLOS ONE

Dear Dr. Bello,

Thank you for submitting your manuscript to PLOS ONE. After careful consideration, we feel that it has merit but does not fully meet PLOS ONE’s publication criteria as it currently stands. Therefore, we invite you to submit a revised version of the manuscript that addresses the points raised during the review process.

We look forward to receiving your revised manuscript.

Kind regards,

Luzia H Carvalho, Ph.D.

Academic Editor

PLOS ONE

Journal Requirements:

2. Please amend your current ethics statement in the Materials and Methods section of your manuscript to include the full name of the ethics committee/institutional review board(s) that approved your specific study.

Additional Editor Comments:

After careful consideration, we felt that your manuscript requires revision, following which it can possibly be reconsidered. Although your manuscript was of interest to the reviewer, major concerns were related to study design and results.  As stated by the reviewers the definition of ASM may be seriously flawed, given that the study population comprised of individuals visiting a hospital due to illness and with malaria parasites. The interpretations of the statistical analyses also need to be revised. In addition, a significant number of issues should be clarified and/or adjust otherwise the MS’s results may be compromised. Finally, we strongly suggest that the MS should be submitted to a copy-editing process. For your guidance, a copy of the reviewers' comments was included below. 

Reviewers' comments:

Reviewer's Responses to Questions

**Comments to the Author**

1. Is the manuscript technically sound, and do the data support the conclusions?

Reviewer #1: Yes

Reviewer #2: No

2. Has the statistical analysis been performed appropriately and rigorously? 

Reviewer #1: Yes

Reviewer #2: No

3. Have the authors made all data underlying the findings in their manuscript fully available?

Reviewer #1: No

Reviewer #2: No

4. Is the manuscript presented in an intelligible fashion and written in standard English?

Reviewer #1: No

Reviewer #2: Yes

5. Review Comments to the Author

Reviewer #1: 1. Title needs modification. Because, it does not much with your objective. So, I recommend your title to be modified as’’ Prevalence of asymptomatic malaria infection and its determinants among older aged patients attending a tertiary hospital in rural Southwestern Nigeria. In addition, what does older aged mean? You have to give clear definition in your operational definition section.

2. In line number 5, you have to write the Authors’ name with superscript first and write their affiliations in increasing order using super scripts. You can give similar super script number for authors at the same affiliation and separate corresponding Author with star mark from other authors. So, remove email, ORCid etc from each. You have to also write all the abbreviations in full terms here.

Lines24 – 27 write as: *Correspondence author: Ibrahim Sebutu BELLO

Email: bello.ibrahim@gmail.com. This one is enough and remove other sentences

3. Abstract:

Here, there is no background section. You have to write the background with statement of the problem in short and show clear message before putting objective.

The objective remains true only if you modify your title as I mentioned earlier.

4. Methods:

Rewrite lines 31-34 clearly including study period, kind of cross sectional design whether community based, health facility based, school based etc.

5. Results: line 39-40, write “being diabetes” as “being diabetic”

6. Conclusion:

Lines 44-45 “The findings may guide ….” is not vital here. But, you have to describe the key findings, for example, prevalence, species and key determinants/factors/ associated.

Introduction

Line49: ref[1] you used is not about ASM. But, it is about worldwide prevalence of anemia. Even it is too old. Please use appropriate citation and remove this.

Methods and materials

Line87: Write Study design area and period as subtitle/ heading2/, then, write like” A hospital based descriptive cross sectional study was carried out at… from …to … (i.e period) ”

What is descriptive cross sectional study design mean?

Line89: the census data (2006) is too old. Can’t you find current/ updated/ data?

Remove line 97. Since, I have mentioned it to write at the beginning of method section.

Line 100: why do these older aged (despite its definition in need?) persons came to hospital? Since they are asymptomatic and your study also is on asymptomatic cases. People with no signs/ symptoms do not attend health institutions. So, how did you get your study subjects in tertiary hospital?

In addition, elderly people especially those above 60 usually remain symptomatic if bitten as they have suppressed immune system. So, how can they remain asymptomatic and be your study subjects. That is why I asked you to operationalize “older aged patients” at the very beginning. I want to hear clear cut of age you used in your study.

Line 104: Also, those … anti-malarial (put treatment) ….

Line 106: sample size… can you clearly show us how you calculated sample size? Here, 16.8% prevalence you used is that obtained by PCR in previous study. So, why did you use PCR value rather than using the value obtained by Microscopy (i.e 7.2%) since the method you conducted to detect ASM is microscopy.

Line 120: remove “The study was conducted between July and October 2021” as I informed you to write in “Study design, area and period section”.

Line 148 – 149: What about to determine P. vivax and other species?

Reference [18] is placed in two areas; lines 143-144 and line 156, which are very far and different ideas. Please write appropriate citation for these ideas each.

Lines 162- 165: According to the national malaria drug policy, …. These sentences are the same and identical with the sentence in lines 143-145, but, with different references. Please see it again and remove either of.

Results

Please describe the species detected!

Line 176: write as “in all, only 19% (44/232)…”

Table1: what were your criteria to classify parasitemia as mild, moderate and severe?

Line 182-184, do you mean the association is in binary logistic regression?

Line 185-187: how do you see the positivity rate of ASM in age groups of 61-80 years old, farmers and informal education? Is that the reality? They have to be symptomatic rather than asymptomatic carriers especially those in 61-80 yrs.

Table 2: in your income classification, is 1.9 dollars daily/weekly/ monthly income? How did you take 1.9 dollar as base line to classify income? Why not 2, 3…or other dollars?

Line 191-193: Do you think that co morbidity does not worsen the signs/symptoms of malaria? For example, why do people with diabetes remain asymptomatic for malaria than non-diabetic?

Line 205: what is window net? Is it physical or chemical barrier? How is it applied?

Table6: here you used Odds ratio as measure of association. But, you applied chi-square in the above. Why do you need to apply two different measures of association? Why couldn’t you use either of one in all?

Discussion

Line 226: put CI in prevalence. B/c the reader can easily understand and compare your value with the values of other studies.

Line 229: reference [21] does not much with the data you used to compare your result. Please see it again.

Line 235-236: However, other studies have disagreed with this report [23, 24]. What was the report of other studies here? You have to specify so that the readers can easily understand. By the way, reference 23 is not relevant and not used to compare your study which is ASM. But, that one is done from febrile/symptomatic/ patients and hence you have to remove.

Line 240: ref [26] can’t be used to compare your study. B/c that study was among children while yours is among older aged. Please note that; in discussion, you have to use the data from the studies among similar study subjects, similar methodology, similar disease condition/asymptomatic only/ etc.

Line 254: …but opposite to several other studies [14, 27]. What do you mean by opposite? Is that lower/ higher? Also please specify the number when you say consistent, higher/ lower in each section so as to make clear for the readers.

Line 259: This may be further aggravated by the low immunity of urban dwellers. Why is the immunity of urban dwellers low?

Line267- 275: In this study, the respondents with mild and moderate anaemia…

This is not your concern and is out of your objective and hence no need to discuss about. Remove it.

Acknowledgement: if this section is needed, why don’t you acknowledge your study participants? They are everything for your study.

General comment: Italicize the genus and species names of parasites/ microorganisms in every section

Reviewer #2: Socio-Demographic Profiles, Co-Morbid Conditions and Preventive Measures as

Determinants of Asymptomatic Malaria Infections Among Older Age Population in

Rural Southwest Nigeria

Review Comments

General: The aim was to assess the prevalence and determinants of asymptomatic infections among individuals 40 years of age and above, resident in a rural community of Ekiti State, Nigeria. The study attempts to present useful information regarding malaria among the elderly, however the definition of ASM may be seriously flawed, given that the study population comprised of individuals visiting a hospital due to illness and with malaria parasites. Thus, the study population can be classified as having clinical malaria and not asymptomatic infection. The interpretations of the statistical analyses are inaccurate. Revision of the language to improve readability, particularly of the introduction sections, using a copy-editing tool, is recommended.

The following minor and major corrections/suggestions could help improve the manuscript:

Title: The authors may consider putting the name of the rural community in the title. The title should indicate that asymptomatic infections were determined by microscopy

Abstract: Overall, there is coherence in abstract but could be improved.

Line 31 to 34: Authors should consider restructuring the sentence to improve readability.

Methods: Authors should consider inclusion of the year of the cross-sectional sampling and the name of the rural community where samples were drawn, in the abstract.

Conclusion line 43: The ‘arguments for high prevalence of asymptomatic malaria in rural settings’, are not shown in results, so the sentence should be clarified.

Introduction: Authors should consider revising the language to improve readability, using a copy-editing tool.

Line 49 to 51: The presence of malaria parasitaemia without symptoms of illness may best be described as an asymptomatic infection, so authors should consider replacing asymptomatic malaria with asymptomatic (malaria) infection.

Line 75 to 78: There are many old-age-associated diseases with socio-economic implications, and planning an effective malaria control programme is not entirely dependent on knowing the prevalence of asymptomatic infections among the elderly. The highest burden of malaria is among children under five years of age, who are the target of several interventional programmes, so the rationale for the study is weak.

Materials and methods:

Line 87: Authors should provide the names of the study institution and the rural community.

Line 97 to 99: Authors should note that an infection identified during a hospital visit will not qualify as asymptomatic.

Line 100: Authors should specify ‘recent history’ (eg. 24 or 48 hours ago or a week ago etc.) in their inclusion criteria/definition of asymptomatic infection

Line 104: Why did Authors exclude individuals with mental illness? Was axillary temperature ≥ 37°C also excluded?

Line 110 to 120: The procedure of the systematic sampling is impressive, however, it is unclear why the sampling is considered random, as the selection of the first respondent appears systematic.

Line 147: May not be relevant.

Line 148, 157: Authors should structure as a minor heading, removing the indent.

Line 161 to 163: Authors should ensure that the referenced treatment guidelines are for the asymptomatic infections (study target population) and not management of clinical malaria.

Authors should provide a clear definition of asymptomatic infection in the methods section.

Results: The authors aim to demonstrate association between several predictor variables and asymptomatic infections, however, the lack of a clear definition for asymptomatic infection (ASM) detracts from the integrity of the data collected. Specifically,

Line 176: Of respondents who tested positive for parasitaemia, how many had symptoms?

Line 179, Table 1: The proportion of individuals testing negative has not been shown to have asymptomatic infections (by PCR) and so should be removed from Table 1, or the heading re-worded. The individuals testing negative should not be placed under the heading ‘Asymptomatic Malaria Parasitaemia’ in the table.

Line 183 to 203: Table 2, 3 and 4 show the distribution of microscopy positive and negative tests by age, sex, occupation etc… and whether the overall differences in the distribution are significant. The tables do not show association between the predictor variables and asymptomatic infections (ASM). Authors should present their results/data properly, for statistical accuracy.

Line 188, Table 2: The reference variables are unclear, if association between the predictors and ASM is the goal. The negative population may or may not have malaria parasites by PCR, so ‘Malaria’ should be replaced with the word ‘Microscopy’.

Line 216, Table 6: The response variable (malaria test positive with or without symptoms = ASM?) for which the regression analyses was performed is missing from the table, so it is difficult to understand how the results were interpreted by the authors.

Discussion: The discussion reads well, based on the results presented. However, due to the lack of a clear definition for ASM and inaccurate interpretations of the statistical analyses, the discussions and conclusions cannot be scientifically supported. The discussion should be re-written after the corrections in the other sections.

6. PLOS authors have the option to publish the peer review history of their article (what does this mean?). If published, this will include your full peer review and any attached files.

Reviewer #1: No

Reviewer #2: No

---

## [Author Response · Author response to Decision Letter 0]

23 Oct 2022

I have uploaded my responses to each of the issues raised by the reviewers

---

## [Decision Letter · Decision Letter 1]

7 Dec 2022

PONE-D-22-24029R1Prevalence Of Asymptomatic Infections By Microscopy And Its Determinants Among Residents of Ido-Ekiti, Southwestern NigeriaPLOS ONE

Dear Dr. Bello,

Thank you for submitting your manuscript to PLoS ONE. After careful consideration, we felt that your manuscript requires substantial revision, following which it can possibly be reconsidered, thus governing the decision of a “major revision”. As requested by the reviewer, the authors still need to address specific issues, particularly related to the data analysis, methods and results. For example, a previous concern on regression analysis section was not addressed. At this point, we strongly recommend that the authors include/clarify all topics raised by the reviewer. For your guidance, a copy of the reviewers' comments was included below.

We look forward to receiving your revised manuscript.

Kind regards,

Luzia H Carvalho, Ph.D.

Academic Editor

PLOS ONE

Reviewers' comments:

Reviewer's Responses to Questions

**Comments to the Author**

1. If the authors have adequately addressed your comments raised in a previous round of review and you feel that this manuscript is now acceptable for publication, you may indicate that here to bypass the “Comments to the Author” section, enter your conflict of interest statement in the “Confidential to Editor” section, and submit your "Accept" recommendation.

Reviewer #1: All comments have been addressed

2. Is the manuscript technically sound, and do the data support the conclusions?

Reviewer #1: Yes

3. Has the statistical analysis been performed appropriately and rigorously? 

Reviewer #1: Yes

4. Have the authors made all data underlying the findings in their manuscript fully available?

Reviewer #1: No

5. Is the manuscript presented in an intelligible fashion and written in standard English?

Reviewer #1: Yes

6. Review Comments to the Author

Reviewer #1: Authors have addressed the previous comments well. But, here are some minor comments and suggestions to make the manuscript better for the readers and scientific community. While responding, It is better if authors give point by point response.

Minor comments:

I think it is good to add the word “malaria” in your title (Prevalence of asymptomatic malaria infection…)

Abstract:

The back ground (Line 23 – 27) is still not adequate. You have to show the gap. I suggest writing as “Asymptomatic infections have received …. Few epidemiological studies on …However; there is limitation on data regarding asymptomatic infections among the old adult populations particularly in the study area. Therefore, this study determined the prevalence of asymptomatic infections by microscopy and its determinants among residents of Ido- Ekiti, Southwestern Nigeria.”

Materials and methods

Authors have put “study population” which is good. But, they also have to write “Source population”

Line 185” Quantitative data were… (Change “are” to were).

Result:

My previous concern on regression analysis section is not addressed. For example, why did you use different measures of association? In tables 3,4 and 5 you used chi-square, But, Odds ratio in table 6. In column, you have to write the number of negatives in each category. You only put number of positives and total. You can leave total, but replace it with negatives. That is why the odds ratios which you put in table 6 are not right through direct manual calculation. Odds ratio remains true if you put Negative column rather than the total making it clear for the readers.

Discussion:

Line 257 – 258: An overall prevalence of asymptomatic P.falciparum infection in this study was 19.0% (95% CI: 14.1% - 24.6%), which is similar to…

Line 264: P.falciparum

Line 275: … 5.298 times more likely to have…

Line 283: Why do type 2 diabetic individuals harbour asymptomatic P. falciparum infection rather than developing symptoms?

Line 296: … five times more likely…

General question: The species detected in your study area was P.falciparum only. How much accurate were you? Who examined the slides? Is the area known by presence of this species only? If not, I expect that there might have been error in species identification in your study.

Finally, align all texts to both the left and right sections. I think all the text is aligned to the left only in your document.

7. PLOS authors have the option to publish the peer review history of their article (what does this mean?). If published, this will include your full peer review and any attached files.

Reviewer #1: No

---

## [Author Response · Author response to Decision Letter 1]

20 Dec 2022

Summary of reviewer’s comments and Authors’ response (PONE-22-24029R1)

Reviewer: 1

comments Response from Author:

Authors have addressed the previous comments well. However, there are minor comments and suggestions to make the manuscript better for the readers and scientific community.

Minor comments:

I think it is good to add the word “malaria” in your title (Prevalence of asymptomatic malaria infection..)

Abstracts:

The background is still not adequate. You have to show the gap. I suggest writing “Asymptomatic infections have received--- Few epidemiological studies on …However; there is limitation on data regarding asymptomatic infections by microscopy and its determinants among residents of Ido-Ekiti, Southwestern Nigeria”

Materials and methods:

Authors have put “ study population” which is good. But, they also have to write” Source population”

Line 185 “Quantitative data are (Change “are” to were)

Results:

My previous concern on regression analysis section is not addressed. For example, why did you use different measures of association? In table 3,4, and 5 you used chi-square, But Odds ratio in table 6. In column, you have to write the number of negatives in each category. You only put number of positive and total. You can leave total, but replace it with negatives. That is why the odds ratios you put in table 6 are not right through direct manual calculation. Odds ratio remains true if you put negative column rather than the total making it clear for the readers.

Discussion:

Line 257-258: An overall prevalence of asymptomatic P. falciparum infection in this study was 19.0% (95% CI: 14.1%-24.6%), which is similar to—

Line 264: P. falciparum

Line 275: --5.298 times more likely to have….

Line 283: Why do type 2 diabetic individuals harbor asymptomatic P. falciparum infection rather than developing symptoms?

Line 296:….. five times more likely……

General questions:

The species detected in your study area was P. falciparum only. How much accurate were you? Who examined the slides? Is the area known by the presence of this species alone? If not, I expect that there might have been error in species identification in your study.

Finally align all texts to both the left and right sections. I think all the text is aligned to the left only in your document.

 Thank you for the efforts put in to review and improve our manuscript.

Line 1: The word “malaria “ has been added to the title. 

Abstract:

Line 21-47: The background has been improved and has addressed the suggestions.

Materials and methods

Line 106-107: Source population has been included.

Line 186: The word “are” has been changed to “were”

Results:

This comment has been addressed. Only one measure of association was now used, which are the odds ratios.

Discussion:

Line 228-229: This has been addressed.

Line 235: This has been addressed

Line 246: The word “as likely “ has been changed to “more likely”

Line 260-264: The association between type 2 diabetic individuals and asymptomatic has been addressed on page line 

Line 271: Five times “as likely” has been changed to “more likely”

There was no error in species identification in this study. P. falciparum has been the only species known to be prevalent in Southwestern Nigeria and this observation has been corroborated by several studies in Southwestern Nigeria. Check references 7,14,35.

The stained slides were taken to the hospital laboratory, where parasitological examinations were made independently by two malaria microscopists, with discrepancies resolved by a senior microscopist who ensured quality control

END.

---

## [Decision Letter · Decision Letter 2]

8 Jan 2023

PONE-D-22-24029R2Prevalence Of Asymptomatic Malaria Infections By Microscopy And Its Determinants Among Residents of Ido-Ekiti, Southwestern NigeriaPLOS ONE

Dear Dr.  Bello,

Thank you for submitting your manuscript to PLoS ONE. After careful consideration, we feel that your manuscript will likely be suitable for publication if the authors revise it to address specific points raised now by the reviewer. According to the reviewer, there are some specific areas where further improvements would be of substantial benefit to the readers.   For your guidance, a copy of the reviewer’s comments was included below.   Please submit your revised manuscript by January 20. If you will need more time than this to complete your revisions, please reply to this message or contact the journal office at plosone@plos.org. Please include the following items when submitting your revised manuscript:A rebuttal letter that responds to each point raised by the academic editor and reviewer(s). You should upload this letter as a separate file labeled 'Response to Reviewers'.A marked-up copy of your manuscript that highlights changes made to the original version. You should upload this as a separate file labeled 'Revised Manuscript with Track Changes'.An unmarked version of your revised paper without tracked changes. You should upload this as a separate file labeled 'Manuscript'.If applicable, we recommend that you deposit your laboratory protocols in protocols.io to enhance the reproducibility of your results. Protocols.io assigns your protocol its own identifier (DOI) so that it can be cited independently in the future. For instructions see: https://journals.plos.org/plosone/s/submission-guidelines#loc-laboratory-protocols. Additionally, PLOS ONE offers an option for publishing peer-reviewed Lab Protocol articles, which describe protocols hosted on protocols.io. Read more information on sharing protocols at https://plos.org/protocols?utm_medium=editorial-email&utm_source=authorletters&utm_campaign=protocols.

We look forward to receiving your revised manuscript.

Kind regards,

Luzia H Carvalho, Ph.D.

Academic Editor

PLOS ONE

Journal Requirements:

Reviewers' comments:

Reviewer's Responses to Questions

**Comments to the Author**

1. If the authors have adequately addressed your comments raised in a previous round of review and you feel that this manuscript is now acceptable for publication, you may indicate that here to bypass the “Comments to the Author” section, enter your conflict of interest statement in the “Confidential to Editor” section, and submit your "Accept" recommendation.

Reviewer #1: All comments have been addressed

2. Is the manuscript technically sound, and do the data support the conclusions?

Reviewer #1: Yes

3. Has the statistical analysis been performed appropriately and rigorously? 

Reviewer #1: Yes

4. Have the authors made all data underlying the findings in their manuscript fully available?

Reviewer #1: No

5. Is the manuscript presented in an intelligible fashion and written in standard English?

Reviewer #1: Yes

6. Review Comments to the Author

Reviewer #1: Authors have addressed the previous concerns well. But, here are additional points to be addressed and included to make manuscript more plausible before publication.

Minor comments:

Line 1: Asymptomatic Malaria Infections (remove s)

Line27: Asymptomatic Malaria Infections (remove s)

Line 60-61: … clinical immunity which is only partially develops … (remove” which is”)

Line 70: Also, the use of…. (Remove also and write as “The use of…)

Line 104- 107:

Source population: This comprised all adult individuals who were residents in Ido-Ekiti, and who presented for the annual free health programs organised by the society of family physicians of Nigeria.

Study population: The study population comprised apparently healthy individuals aged 40 years and above.

Line 108-111: Consented apparently healthy individuals …dizziness, joint pain, anorexia and malaise were included in the study.

Line 112-14: Individuals who were …conduct of this study were excluded.

Line 188: Italicize “Plasmodium”

7. PLOS authors have the option to publish the peer review history of their article (what does this mean?). If published, this will include your full peer review and any attached files.

Reviewer #1: No

---

## [Author Response · Author response to Decision Letter 2]

9 Jan 2023

I have addressed all the specific issues raised in the reviewers comment for the author

---

## [Editor Report · Decision Letter 3]

12 Jan 2023

Prevalence Of Asymptomatic Malaria Infection By Microscopy And Its Determinants Among Residents of Ido-Ekiti, Southwestern Nigeria

PONE-D-22-24029R3

Dear Dr.Bello,

We’re pleased to inform you that your manuscript has been judged scientifically suitable for publication and will be formally accepted for publication once it meets all outstanding technical requirements.

Kind regards,

Luzia H Carvalho, Ph.D.

Academic Editor

PLOS ONE
---

## [Editor Report · Acceptance letter]

3 Feb 2023

PONE-D-22-24029R3 

Prevalence Of Asymptomatic Malaria Infection By Microscopy And Its Determinants Among Residents of Ido-Ekiti, Southwestern Nigeria 

Dear Dr. Bello:

I'm pleased to inform you that your manuscript has been deemed suitable for publication in PLOS ONE. Congratulations! Your manuscript is now with our production department. 

Kind regards, 

on behalf of

Dr. Luzia H Carvalho 

Academic Editor

PLOS ONE